# A2A: Mechanistic Analysis for Efficient Layer Selection in Activation Steering

## Abstract

Activation steering has emerged as an effective and economical technique for behavior control in large language models (LLMs). Despite growing interest, existing methods typically rely on exhaustive layer-wise interventions to identify effective steering locations. This process is computationally expensive and lacks interpretability. To mitigate this problem, we propose Attribution-to-Action (A2A), an efficient layer selection framework that leverages mechanistic interpretability to identify layers where steering is most impactful. Specifically, A2A first constructs an attribution graph that traces how internal pathways contribute to model outputs, guided by a small set of contrastive behavior data. Subsequently, edge-level attribution weights are aggregated at the node level and then combined within each layer to derive an importance ranking. Steering vectors are applied only to the top-ranked layers, effectively reducing the search space. Experiments on behavior control tasks such as personality conditioning and model jailbreaking demonstrate that A2A achieves performance comparable to exhaustive search while requiring significantly fewer interventions and offering improved interpretability.

## 1 Introduction

Large language models (LLMs) have achieved remarkable progress in reasoning, dialogue, and personalization (Yang et al., 2025; Shao et al., 2024; Zhou et al., 2024). As these systems are deployed in practical applications, the ability to control their behaviors and personalities has become increasingly important. Steering vectors have emerged as an efficient and effective technique for this purpose, requiring no fine-tuning. By injecting a learned direction into the residual stream of a model at a particular layer with a chosen scaling factor, LLMs can be encouraged to display or suppress specific behaviors. This approach has been applied to various downstream tasks, such as reducing hallucinations (Su et al., 2025), avoiding sycophancy (Siddique et al., 2025), enforcing refusals (Ghosh et al., 2025), and simulating personality traits (Chen et al., 2025).

However, the effectiveness of steering strongly depends on the choice of intervention layer. The prevailing approach is exhaustive grid search across all layers and multiple scaling values (Panickssery et al., 2023; Turner et al., 2023; Hao et al., 2025). Although this method can identify the most effective locations, it is computationally expensive and scales poorly with larger models. To reduce computational cost, some works limit the search to a narrow set of layers, usually the middle layers, based on empirical evidence (Konen et al., 2024). These shortcuts may yield reasonable results but risk missing the true points of influence, and they provide little understanding of why certain layers matter.

Mechanistic interpretability has been proposed as a principled lens for understanding model behaviors. Prior studies construct attribution graphs and retain only the necessary subgraphs, known as circuits, showing that as little as 10% of a model can be sufficient to reproduce performance on certain tasks (Elhage et al., 2021; Olsson et al., 2022; Yao et al., 2024). This suggests that interventions need not search blindly across all layers but can instead target those with the greatest influence on a given behavior. Such a strategy reduces computational cost and clarifies why particular layers are effective for steering. Motivated by this insight, our work leverages mechanistic interpretability to devise a layer selection method for activation steering.

We propose Attribution-to-Action (A2A), a framework that aggregates edge attributions into layer-importance rankings (see Figure 1). For each behavior, we compute incoming, outgoing, and total

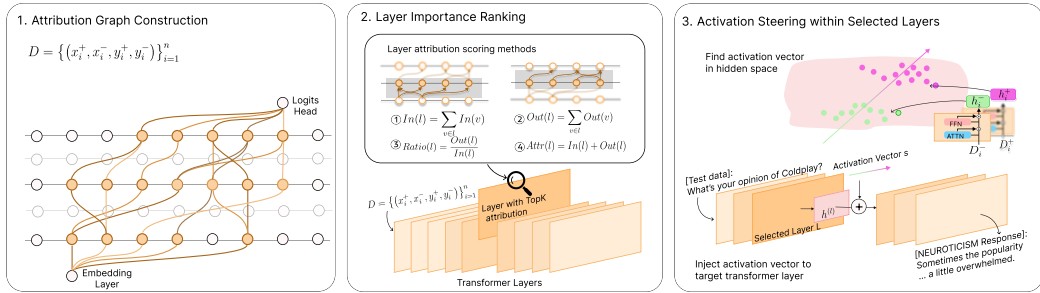

Figure 1: Overview of the A2A framework. Given a contrastive dataset, we first construct an attribution graph using EAP-IG (*Attribution Graph Construction*). Edge attributions are then aggregated into layer-level importance scores with four scoring methods: In, Out, Total and Ratio (*Layer Importance Ranking*). Finally, the search space is limited to the top-$k$ layers, where an activation vector $s$ is injected into the residual stream (*Steering with A2A-Selected Layers*). This procedure reduces the cost of layer selection compared to exhaustive search while preserving strong task performance.

contributions per layer, along with contrastive measures such as ratios to capture directional flow. With only modest computation, A2A yields a ranked list of candidate layers for steering, thereby limiting the search space to the top-$k$ layers rather than exhaustively evaluating all layers. We evaluate A2A on two behavioral control tasks, namely personality steering and alignment hacking. Experiments on Llama2-7B-Chat, Llama2-13B-Chat, and Qwen2.5-7B-Instruct show that A2A achieves close alignment with exhaustive grid search, as measured by nDCG for ranking consistency and task accuracy under steering, while reducing computation and maintaining near-optimal performance.

Our contributions are summarized as follows:

- We reformulate layer selection for activation steering from a mechanistic interpretability perspective, showing that layer importance derived from attribution graphs is highly correlated with the layers where steering interventions are most effective (see Section 3.5).

- We propose *Attribution-to-Action (A2A)*, a framework that aggregates edge-level attributions into layer-importance rankings. By computing incoming, outgoing, and composite measures with only modest computation, A2A produces a ranked list of candidate layers for steering without resorting to exhaustive grid search.

- We demonstrate the effectiveness and generality of A2A through extensive experiments on personality steering and alignment hacking tasks, using the PersonalityEdit and SafeEdit datasets. Results on Llama2-7B-Chat, Llama2-13B-Chat and Qwen2.5-7B-Instruct show that A2A achieves ranking consistency close to exhaustive search (measured by nDCG) while maintaining near-optimal task accuracy with lower computational cost (see Section 4.1).

## 2 ATTRIBUTION-TO-ACTION (A2A)

We introduce **Attribution-to-Action (A2A)**, an efficient layer selection framework for activation steering in large language models (LLMs). Rather than exhaustively sweeping across all layers, A2A leverages mechanistic analysis to identify where interventions are most likely to be effective. It consists of three steps: (1) constructing an attribution graph that captures how internal activations propagate and influence model outputs; (2) computing layer importance via aggregated edge contributions; (3) using the resulting layer ranking to restrict the search space, applying steering vectors one layer at a time within the top-$k$ candidates.

The remainder of this section is organized as follows. We first introduce the attribution mechanism (§3.1), then present our layer selection algorithm (§3.2), and finally describe how steering is performed (§3.3).

## 2.1 ATTRIBUTION GRAPH CONSTRUCTION

In mechanistic interpretability, neural networks can be represented as directed acyclic graphs where each node corresponds to a specific computational unit, such as an input embedding, an attention head, or an MLP block (Olah et al., 2020; Wang et al., 2022; Conmy et al., 2023b; Yao et al., 2024; Lindsey et al., 2025). The edges represent interactions between these components, including residual connections and projection operations. In the initial step of **A2A**, we construct an attribution graph $\mathcal{G} = (N, E)$ to capture how internal components contribute to the model's output. For transformer decoders, the node set $N$ includes input embeddings $I$, output unembeddings $O$, attention heads $A_l$, and MLP blocks $M_l$ at each layer $l$. Each directed edge $(u, v) \in E$ represents the flow of activation-level influence from component $u$ to component $v$, with both $u, v \in N$. This graph provides a structured abstraction of how information flows through the model and serves as the foundation for our subsequent attribution-based computation of edge, node, and layer importance.

To quantify the influence of internal components on specific behavioral outcomes, we adopt the attribution method known as Edge Attribution Patching with Integrated Gradients (EAP-IG), proposed by Hanna et al. (2024b). This method requires a contrastive input pair $(x^+, x^-)$ and a corresponding output pair $(y^+, y^-)$, where $x^+$ is a clean input designed to elicit a target behavior (e.g., high Openness or a jailbreak), and $x^-$ is a semantically similar, corrupted input that does not induce the behavior. Likewise, $y^+$ and $y^-$ are contrasting model responses—one aligned with the behavior and the other not.

For each edge $(u, v)$ in the attribution graph $\mathcal{G}$, The attribution score quantifies the influence of source node activations on target node activations and their contribution to the model's preference between the two contrastive outputs. Let $z_u^+$ and $z_u^-$ denote the activations at node $u$ under inputs $x^+$ and $x^-$, respectively. Let $z_v$ represent the downstream activation at node $v$, which receives input from $u$. The attribution is computed as:

$$\text{Attr}(u \to v) = (z_u' - z_u) \cdot \frac{1}{m} \sum_{k=1}^{m} \frac{\partial L\left(z_u' + \frac{k}{m}(z_u - z_u')\right)}{\partial z_v} \tag{1}$$

After two forward passes and one backward pass, each edge in the attribution graph is assigned a score indicating its importance. In prior work on mechanistic interpretability, a common next step is to prune low-scoring edges from the attribution graph to extract a minimal subgraph sufficient for performing the target task (Yao et al., 2024; Ou et al., 2025; Dunefsky et al., 2024; Tigges et al., 2024). In contrast, we retain the full attribution graph without pruning, as our objective is not to identify a minimal generative circuit, but rather to preserve all node-to-node influence signals for subsequent node-level aggregation and layer importance ranking.

## 2.2 LAYER IMPORTANCE RANKING

Given the attribution graph $\mathcal{G} = (N, E)$ with edge weights derived in Section 2.1, our goal is to identify layers that play central roles in propagating behavioral influence. To this end, we perform a multi-level aggregation of edge attributions, first at the node level and then at the layer level.

**Node-Level Attribution** We define the influence of each node $v \in N$ based on the attribution flow it receives and transmits through incoming and outgoing edges:

$$\text{In}(v) = \sum_{(u \to v) \in E} |\text{Attr}(u \to v)|, \qquad \text{Out}(v) = \sum_{(v \to w) \in E} |\text{Attr}(v \to w)| \tag{2}$$

Here, $\text{In}(v)$ captures how much node $v$ is influenced by upstream components, while $\text{Out}(v)$ quantifies how much $v$ influences downstream components. Together, they reflect the directional flow of behavioral influence through the node.

**Layer-Level Attribution** Next, we group all nodes by their associated layer indices and aggregate node-level attribution scores to obtain layer-level importance. For each layer $l$, we compute:

- **In Attribution:** $\text{Attr}(l) = \sum_{v \in l} \text{In}(v)$
  Measures the total incoming attribution to all nodes in layer $l$.

- **Out Attribution:** $\text{Attr}(l) = \sum_{v \in l} \text{Out}(v)$
  Measures how much influence layer $l$ exerts on subsequent layers.

- **Total Attribution:** $\text{Attr}(l) = \text{In}(l) + \text{Out}(l)$
  Reflects the overall participation of layer $l$ in the attribution flow.

- **Out-to-In Ratio:** $\text{Attr}(l) = \frac{\text{Out}(l)}{\text{In}(l) + \epsilon}$
  Highlights the relative dominance of layer $l$ as an influencer in attribution propagation.

**Layer Importance Ranking**  After computing all layer-level scores, we rank the layers in descending order according to a chosen metric (e.g., total attribution or out-to-in ratio). To reduce the computational burden of activation steering, we perform intervention only within the top-$k$ most important layers, rather than sweeping across all layers.

Given a contrastive dataset for a target behavior, Algorithm 1 summarizes the procedure for identifying the layers with the greatest influence. The algorithm aggregates edge-level attributions into a layer-level score dictionary $\mathcal{S}$ and returns the top-$k$ ranked layers.

---

**Algorithm 1** A2A: Attribution-to-Action Layer Selection

---

**Input:**
   Contrastive dataset $\mathcal{D} = \{(x_i^+, x_i^-, y_i^+, y_i^-)\}_{i=1}^n$;
   Zero-initialized attribution graph $\mathcal{G} = (N, E)$;
   Layer scoring method $\mathcal{M} \in \{\text{In}, \text{Out}, \text{Ratio}, \text{Total}\}$;
   Number of top layers $k$
**Output:** Top-$k$ most influential layers for activation steering

1: **for all** $(x^+, x^-, y^+, y^-) \in \mathcal{D}$ **do**          $\triangleright$ Build attribution graph on behavioral dataset
2:     Compute edge attributions $\text{Attr}(u \to v)$ for all $(u \to v) \in E$
3:     Accumulate: $\mathcal{G}[u \to v] \mathrel{+}= |\text{Attr}(u \to v)|$
4: **end for**
5: Initialize node-level statistics $\text{In}(v), \text{Out}(v)$ for all $v \in N$
6: **for all** edge $(u \to v) \in \mathcal{G}$ **do**          $\triangleright$ Aggregate edge attributions into node statistics
7:     $w \leftarrow \mathcal{G}[u \to v]$
8:     $\text{Out}(u) \mathrel{+}= w, \quad \text{In}(v) \mathrel{+}= w$
9: **end for**
10: Initialize layer importance dictionary $\mathcal{S} \leftarrow \{\}$
11: **for all** layer $l$ **do**          $\triangleright$ Aggregate node statistics into layer scores
12:     **if** $\mathcal{M} = \text{In}$ **then**
13:         $\mathcal{S}[l] \leftarrow \sum_{v \in l} \text{In}(v)$
14:     **else if** $\mathcal{M} = \text{Out}$ **then**
15:         $\mathcal{S}[l] \leftarrow \sum_{v \in l} \text{Out}(v)$
16:     **else if** $\mathcal{M} = \text{Ratio}$ **then**
17:         $\mathcal{S}[l] \leftarrow \frac{\sum_{v \in l} \text{Out}(v)}{\sum_{v \in l} \text{In}(v) + \epsilon}$
18:     **else if** $\mathcal{M} = \text{Total}$ **then**
19:         $\mathcal{S}[l] \leftarrow \sum_{v \in l} (\text{In}(v) + \text{Out}(v))$
20:     **end if**
21: **end for**
22: Sort layers by $\mathcal{S}[\cdot]$ in descending order          $\triangleright$ Rank layers and select top-$k$
23: **return** Top-$k$ layers

---

## 2.3 STEERING WITH A2A-SELECTED LAYERS

Once the top-$k$ layers are identified, we apply **activation steering** only at these layers. Given a steering vector $s \in \mathbb{R}^d$ and scaling factor $\lambda$, we modify the model's hidden state at layer $l$ by:

$$h_i^{(l)} \leftarrow h_i^{(l)} + \lambda \cdot s \tag{3}$$

Compared to full grid search over all layers, the search space is limited to the top-ranked layers, which reduces computation while maintaining effective steering.

# 3 EXPERIMENTS

We evaluate **A2A (Attribution-to-Action)** across two contrasting behavior control scenarios: *personality steering*, a stylistic generation task, and *Jailbreaking*, a decision-control task that aims to elicit unsafe responses from a well-aligned LLM. These two tasks differ in direction and difficulty, providing a comprehensive testbed for the robustness and generality of layer selection strategies.

## 3.1 TASK SETUP

We evaluate A2A on two distinct behavior control tasks: *personality steering*, which targets stylistic modulation of model outputs, and *alignment hacking*, which aims to elicit unsafe responses despite the model's alignment. These tasks are instantiated with the PersonalityEdit and SafeEdit datasets, respectively.

**Personality Steering.** We adopt the PersonalityEdit dataset (Mao et al., 2024), which is designed to evaluate controllable generation across three representative Big Five personality traits: Extraversion, Agreeableness, and Neuroticism. Each instance consists of a dialogue context, an associated entity, and multiple responses labeled with target traits. To construct contrastive data for attribution analysis, we treat trait-consistent prompts as positive samples and those aligned with other traits as negative samples, pairing them with their corresponding responses to form $(x^+, x^-, y^+, y^-)$ tuples.

**Alignment Hacking.** We further experiment with the SafeEdit dataset (Wang et al., 2024), which provides paired harmful prompts and completions $(x_{\text{harm}}, y_{\text{harm}})$ as well as safe counterparts $(x_{\text{safe}}, y_{\text{safe}})$. We repurpose these pairs to construct contrastive datasets by treating safe and unsafe examples as positive and negative samples. While the dataset was originally introduced for editing models toward safer behaviors, the high safety alignment of modern LLMs makes this setting relatively easy. To create a more challenging variant, we adopt *jailbreaking*, where the objective is to steer the model away from safe completions toward unsafe generations, and we evaluate success rates under this adversarial setting.

## 3.2 EVALUATION METRICS

We evaluate A2A from two perspectives: (1) the consistency between its predicted layer rankings and ground-truth rankings obtained via exhaustive search, and (2) the task performance under different layer selection strategies.

**Ranking Consistency.** We also evaluate the consistency between A2A's predicted layer rankings and the ground-truth rankings obtained through exhaustive layer-wise search. To this end, we report normalized discounted cumulative gain (NDCG) at different cutoff points $k$, which measures how well the top-ranked layers identified by A2A align with the most effective layers discovered by exhaustive search. We adopt NDCG@1, NDCG@5, and NDCG@10 to capture agreement at different granularities, from identifying the single most important layer to recovering a broader set of influential layers.

**Task Performance.** We evaluate task performance under different layer selection strategies. To ensure reproducibility, we use the official classifiers provided with each dataset. For PersonalityEdit, accuracy is measured by whether generated responses are classified into the intended Big Five trait by the released trait classifier. For SafeEdit, we report the proportion of completions classified as unsafe by the official safety classifier, which reflects the success rate of jailbreaking under reverse steering.

## 3.3 BACKBONE MODELS.

We evaluate A2A on LLaMA2-7B-Chat, LLaMA2-13B-Chat, and Qwen2.5-7B-Instruct to assess its generalization across both model sizes and architectures. We select these models because they are widely adopted in the research community and have demonstrated strong performance on a broad range of language tasks.

Table 1: NDCG@$k$ results on personality steering tasks. Higher is better. Grid Search is reported as an upper bound baseline, since it exhaustively evaluates all layers to obtain the optimal ranking but at prohibitive computational cost. The best scores in each column are highlighted in bold.

| Method | Extraversion | | | Agreeableness | | | Neuroticism | | | Average |
|---|---|---|---|---|---|---|---|---|---|---|
| | @1 | @5 | @10 | @1 | @5 | @10 | @1 | @5 | @10 | |
| Grid Search | 1.00 | 1.00 | 1.00 | 1.00 | 1.00 | 1.00 | 1.00 | 1.00 | 1.00 | 1.00 |
| Random | 0.03 | 0.07 | 0.13 | 0.00 | 0.18 | 0.20 | 0.13 | 0.12 | 0.09 | 0.11 |
| ActDiff | 0.35 | 0.70 | 0.72 | 0.80 | 0.88 | 0.89 | 0.42 | 0.76 | 0.78 | 0.70 |
| **A2A** | | | | | | | | | | |
| ├ A2A-In | 0.12 | 0.62 | 0.62 | 0.12 | 0.57 | 0.58 | 0.12 | 0.64 | 0.65 | 0.45 |
| ├ A2A-Out | **0.50** | 0.84 | 0.87 | **1.00** | **1.00** | **1.00** | 0.50 | 0.88 | 0.88 | **0.83** |
| ├ A2A-Total | **0.50** | **0.88** | **0.88** | **1.00** | 0.99 | **1.00** | 0.50 | **0.88** | **0.88** | **0.83** |
| └ A2A-Ratio | **0.50** | 0.35 | 0.62 | **1.00** | 0.94 | 0.93 | **1.00** | 0.94 | 0.93 | 0.80 |

## 3.4 EXPERIMENTAL COMPARISONS

We compare the following layer selection strategies:

- **Grid Search**: Exhaustively applies steering at all layers to identify the best-performing ones, serving as an upper bound.
- **Random**: For the Random baseline, we generate 10 random permutations of layers and report the average results.
- **Mid-Layer Search**: Following prior work on style steering (Konen et al., 2024), steering is restricted to layers 18–20 based on empirical effectiveness.
- **Activation Difference (ActDiff)**: Follow interpretability studies (Tigges et al., 2024), which identify important components using activation differences. We adapt this approach to select influential layers.
- **Attribution-to-Action (A2A)**: Our proposed framework, with four variants based on different aggregation strategies: **A2A-In**, **A2A-Out**, **A2A-Total**, and **A2A-Ratio**.

## 3.5 MAIN RESULTS

Table 1 reports NDCG@$k$ on the Personality Steering task. As expected, *Grid Search* achieves perfect scores across all traits, serving as an upper bound but requiring exhaustive evaluation over all layers. Random selection performs poorly, highlighting the difficulty of the task. The heuristic *ActDiff* baseline substantially improves over random, confirming that activation differences carry useful signals for layer relevance, but it still lags behind our approach. Among the A2A variants, A2A-Out and A2A-Total consistently achieve the best performance, matching or closely approaching the *Grid Search* upper bound with far fewer interventions. This demonstrates that attribution-based aggregation effectively captures the layers most responsible for controlling personality expression, yielding accurate rankings without the computational overhead of exhaustive search.

Table 2: NDCG@$k$ for alignment hacking. Higher is better. Grid Search is an upper bound via exhaustive search. Column bests are **bold**.

| Method | Jailbreak | | | Average |
|---|---|---|---|---|
| | @1 | @5 | @10 | |
| Grid Search | 1.00 | 1.00 | 1.00 | 1.00 |
| Random | 0.09 | 0.15 | 0.18 | 0.14 |
| ActDiff | 0.42 | 0.70 | 0.74 | 0.62 |
| **A2A Variants** | | | | |
| ├ A2A-In | 0.13 | 0.71 | 0.76 | 0.53 |
| ├ A2A-Out | **1.00** | 0.77 | **0.79** | 0.85 |
| ├ A2A-Total | **1.00** | **0.78** | **0.79** | **0.86** |
| └ A2A-Ratio | **1.00** | 0.75 | 0.76 | 0.84 |

To evaluate the generality of our method beyond stylistic generation, we further test it on the Alignment Hacking task, which targets behavioral safety alignment. As shown in Table 2, *Grid Search* again provides an ideal upper bound, while random selection remains ineffective. The ActDiff baseline yields moderate gains, but its performance plateaus, suggesting that raw activation magnitude alone may not suffice under adversarial settings.

Table 3: Accuracy (%) after activation steering on personality steering and jailbreak tasks. Higher is better.

| Method | Ext | Agr | Neu | Pers-Avg | Jail |
|---|---|---|---|---|---|
| Grid Search | 100 | 92 | 85 | 92 | 64 |
| Random | 52 | 52 | 51 | 52 | 15 |
| Mid-Layer | 64 | 75 | **85** | 75 | 44 |
| ActDiff | 64 | 64 | 65 | 64 | 25 |
| **A2A Variants** | | | | | |
| └ A2A-In | **100** | 76 | 84 | **87** | 56 |
| ├ A2A-Out | **100** | 85 | 69 | 85 | 49 |
| ├ A2A-Total | **100** | **92** | 69 | **87** | **64** |
| └ A2A-Ratio | **100** | 74 | 84 | 86 | 44 |

In contrast, A2A variants sustain strong performance, with A2A-Out and A2A-Total achieving the highest averages (0.86 and 0.85, respectively), reaffirming their effectiveness across distinct steering objectives. Notably, A2A-Total attains the strongest results, indicating that combining incoming and outgoing attributions provides complementary signals. This aggregation helps identify genuinely influential layers more reliably than considering either direction alone. A2A-In, by comparison, continues to underperform, underscoring the importance of capturing output effects in the attribution process.

Importantly, A2A variants achieve performance close to Grid Search while requiring less computation than exhaustive layer-wise search. These findings establish A2A as a robust and general framework for efficient layer selection in diverse activation steering scenarios.

## 3.6 TASK ACCURACY

To assess whether A2A can lead to effective behavioral control, we report task accuracy after activation steering in Table 3. As expected, Grid Search serves as the upper bound, achieving the highest accuracy across both personality steering (with 92% on average) and jailbreak (64%) tasks. Random layer selection yields the weakest performance, reflecting the difficulty of locating effective intervention points without guidance. The Mid-Layer baseline improves over random, particularly on Neuroticism, but its fixed selection strategy lacks flexibility. ActDiff further boosts performance, suggesting that activation differences can partially reflect relevant features, though it still underperforms on the Jailbreak task.

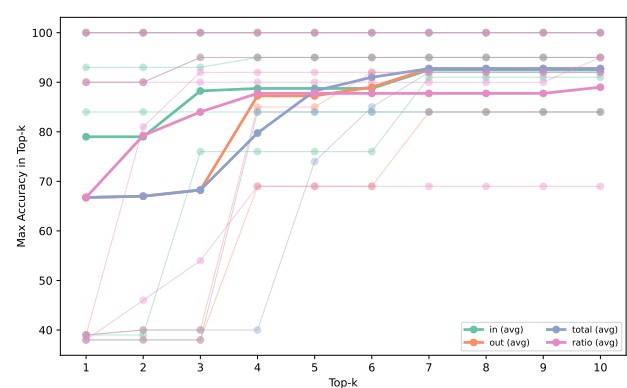

Figure 2: Task accuracy with varying $k$ in A2A. Performance increases with $k$ and plateaus after $k = 10$. We adopt $k = 5$ as a balanced choice between accuracy and cost.

A2A variants consistently outperform all baselines and are most comparable to the upper bound provided by Grid Search. Notably, A2A-In and A2A-Total match Grid Search on personality traits like Extraversion and Agreeableness, with A2A-Total achieving the highest overall alignment (87%) and the best jailbreak accuracy (64%). This suggests that aggregating bidirectional attribution (both incoming and outgoing) provides a more reliable signal for steering. A2A-Ratio and A2A-Out also show strong performance, highlighting the robustness of A2A under different aggregation schemes. The results confirm that A2A not only approximates exhaustive search in ranking but also yields high-quality behavioral control in practice.

## 4 DISCUSSION

### 4.1 COMPUTATIONAL COST ANALYSIS

We compare the computational complexity of A2A with exhaustive Grid Search. Grid Search requires evaluating steering at every layer on the evaluation set, with cost $\text{Cost}_{\text{grid}} \approx L \cdot |\mathcal{E}|$, which grows linearly with both the number of layers $L$ and evaluation set size $|\mathcal{E}|$. In contrast, A2A constructs dataset-level attribution graphs using EAP-IG with $m = 5$ integration steps. Each attribution requires 5 forward and 5 backward passes, where a backward is approximately two forwards, yielding about 15 forward-equivalents per sample. This attribution stage costs $15 \cdot |\mathcal{D}_{\text{attr}}|$, where $|\mathcal{D}_{\text{attr}}|$ is the attribution dataset size. After ranking, steering is applied only to the top-$k$ layers, adding $k \cdot |\mathcal{E}|$. Thus the total cost of A2A is

$$\text{Cost}_{\text{A2A}} \approx 15 \cdot |\mathcal{D}_{\text{attr}}| + k \cdot |\mathcal{E}|.$$

Since $k \ll L$ and $|\mathcal{D}_{\text{attr}}|$ is modest (a few hundred samples in our experiments), A2A achieves near-optimal layer selection at a fraction of the cost of grid search.

For instance, on the SafeEdit validation set with 2.7k test examples, Grid Search must perform steering at all 32 layers, resulting in 86,400 total forward passes. In contrast, A2A requires only 19,500 forward-equivalent steps. This amounts to a 77.4% reduction in computation while maintaining comparable steering performance. These results empirically validate A2A as an efficient and scalable alternative to exhaustive layer-wise search.

### 4.2 EFFECT OF TOP-$k$ SELECTION

A2A ranks all layers by their attribution scores and restricts steering to the top-$k$ candidates. Increasing $k$ expands the candidate set, providing a higher chance of including the most effective layers and thus improving task performance, but it also raises computational cost. As shown in Figure 2, performance increases steadily with $k$ and plateaus around $k = 10$, where near-optimal accuracy is achieved. In practice, we set $k = 5$, which already provides strong performance while keeping computation low.

Figure 3: Task accuracy with varying $k$ in A2A. Performance increases with $k$ and plateaus after $k = 10$. We adopt $k = 5$ as a balanced choice between accuracy and cost.

### 4.3 EFFECT OF ATTRIBUTION DATASET SIZE

Another important factor in the scalability of A2A is the number of samples used to construct the attribution graph. Since attribution requires multiple forward and backward passes, the dataset size directly determines the computational overhead. Figure 3 shows the relationship between dataset size and the average NDCG@5 across all tasks. We observe that ranking quality improves rapidly as the dataset size increases from 100 to 200 examples and stabilizes around 400 examples, beyond which further gains are marginal. This indicates that only a modest number of contrastive pairs is sufficient to produce reliable layer rankings, reflecting the concentration of attribution flows. Therefore, we fix the attribution dataset size to 400 in all main experiments to balance efficiency and effectiveness.

### 4.4 GENERALIZATION ANALYSIS

We further evaluate the generalizability of A2A across different model sizes and architectures, with results reported in Table 4 for LLaMA2-7B-Chat, LLaMA2-13B-Chat, and Qwen2.5-7B-Instruct. Across all settings, A2A variants consistently outperform baselines such as Random, Mid-Layer,

and ActDiff, often approaching the performance of exhaustive Grid Search while requiring substantially fewer computations. Importantly, A2A maintains strong performance on larger models (LLaMA2-13B) and on models trained with different alignment objectives (Qwen2.5). For example, in personality steering tasks (PA), A2A-Total achieves accuracies of 87%, 89%, and 90% on the three models respectively, closely matching Grid Search. Similarly, in jailbreak tasks, A2A-In and A2A-Total provide the largest improvements, highlighting the robustness of our method under safety-critical settings. These results demonstrate that A2A does not rely on model-specific heuristics but generalizes effectively across scales and architectures, making it a practical solution for steering in diverse LLM families.

## 5 RELATED WORK

**Mechanistic Interpretability**   Mechanistic interpretability has been proposed to reverse-engineer model behaviors into *circuits*, which are small subgraphs of attention heads, MLPs, and their connections that are causally sufficient for a behavior (Elhage et al., 2021; Olsson et al., 2022). A standard analysis method is intervention-based testing, where the model is run on a corrupted input and internal activations are selectively restored from a clean run to localize causal pathways associated with specific behaviors (Goldowsky-Dill et al., 2023). However, per-edge interventions do not scale to large models. To address this, gradient-based methods have been proposed that approximate causal effects with far fewer runs, such as ACDC (Conmy et al., 2023a) and attribution patching (Syed et al., 2023). However, raw gradient signals are often noisy and may not reliably reflect true causal impact. Recent work mitigates this issue by integrating gradients along interpolation paths and proposing Edge Attribution Patching with Integrated Gradients (EAP–IG) (Hanna et al., 2024a). In parallel, theoretical work on the *linear representation hypothesis* formalizes when high-level concepts correspond to linear directions in representation space, linking probing, steering vectors, and counterfactual interventions (Park et al., 2023). Taken together, these findings suggest that behaviors are routed through sparse, layered flows concentrated in the residual stream, motivating layer-aware analysis. In this work, rather than pruning the attribution graph into sparse circuits, we retain the full graph and aggregate attribution signals at the layer level as the basis for our analysis.

**Activation Steering in Large Language Models**   Activation steering modifies hidden states at inference time to effectively control model behaviors without finetuning. Early work computes *activation additions* by contrasting activations from paired prompts and injecting the difference as a steering vector (Turner et al., 2023). *Contrastive Activation Addition* (CAA) averages differences over a dataset of contrastive pairs and applies the resulting vector across positions to steer properties such as truthfulness, toxicity, or sycophancy, with minimal capability loss (Panickssery et al., 2023). *Inference-Time Intervention* (ITI) learns directions over a small set of attention heads to elicit truthful answers on TruthfulQA, trading off helpfulness via a strength parameter (Li et al., 2023). Subsequent variants steer via alternative bases (e.g., latent/task-specific or spectral spaces) or learned probes, improving specificity and compositional control (Liu et al., 2023; Qiu et al., 2024; Chen et al., 2024).However, most methods either (i) inject at a fixed or canonical depth, (ii) spread uniformly across many layers, or (iii) rely on costly grid search over layers and scales. These approaches offer limited interpretability regarding *why* certain layers are effective. In contrast, we ground layer selection in mechanistic interpretability, estimating per-layer contributions from an attribution graph and steering only at the top-$k$ layers. This reframes the question of "where to steer" from a heuristic or exhaustive search problem into an interpretable, data-driven decision rule that preserves steering efficacy while reducing computational cost.

## 6 CONCLUSION

In this paper, we propose Attribution-to-Action (A2A), an efficient method for selecting effective layers for activation steering. A2A leverages mechanistic interpretability to rank layers by attribution scores and applies steering only at the top candidates, avoiding costly grid search. Experiments show that A2A achieves ranking consistency close to exhaustive search and maintains near-optimal task accuracy with much lower computation. Our study reveals that attribution flows in the graph align closely with layers most effective for intervention, bridging activation steering and mechanistic interpretability.

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

# 7 APPENDIX

## 7.1 DISCLOSURE OF LLM USAGE

Large language models (LLMs) were used exclusively for text polishing and translation. They were not involved in the design of the methodology, implementation of experiments, analysis of results, or writing of scientific content. The authors take full responsibility for all claims, results, and conclusions presented in this paper.

Table 4: Generalizability evaluation of A2A given by LLaMA2-7B-Chat, LLaMA2-13B-Chat, and Qwen2.5-7B-Instruct on personality tasks (E for Extraversion, A for Agreeableness, N for Neuroticism, PA for average personality accuracy) and jailbreak tasks (J)

| Method | LLaMA2-7B-Chat | | | | | LLaMA2-13B-Chat | | | | | Qwen2.5-7B-Instruct | | | | |
|---|---|---|---|---|---|---|---|---|---|---|---|---|---|---|---|
| | E | A | N | PA | J | E | A | N | PA | J | E | A | N | PA | J |
| Grid Search | 100 | 92 | 85 | 92 | 64 | 97 | 96 | 90 | 94 | 74 | 97 | 94 | 90 | 94 | 90 |
| Random | 52 | 52 | 51 | 52 | 15 | 55 | 50 | 52 | 52 | 25 | 55 | 52 | 47 | 51 | 32 |
| Mid-Layer | 64 | 75 | **85** | 75 | 44 | 68 | 75 | 79 | 74 | 58 | 64 | 74 | 79 | 72 | 63 |
| ActDiff | 64 | 64 | 65 | 64 | 25 | 69 | 66 | 74 | 70 | 42 | 62 | 73 | 65 | 67 | **78** |
| **A2A Variants** | | | | | | | | | | | | | | | |
| ⊢ A2A-In | **100** | 76 | 84 | **87** | **56** | 94 | 88 | 85 | 89 | **72** | 96 | 85 | 80 | 87 | 56 |
| ⊢ A2A-Out | **100** | 85 | 69 | 85 | 49 | **97** | 90 | 75 | 87 | 62 | 96 | 92 | 78 | 89 | 75 |
| ⊢ A2A-Total | **100** | 92 | 69 | **87** | 64 | 95 | **94** | 71 | 87 | 64 | **97** | **93** | 81 | **90** | 77 |
| ⊢ A2A-Ratio | **100** | 74 | 84 | 86 | 44 | 91 | 84 | **86** | 87 | 60 | 95 | 85 | **85** | 88 | 72 |

