# OpenReview forum: "A2A: Mechanistic Analysis for Efficient Layer Selection in Activation Steering"
_ICLR.cc/2026/Conference — ICLR 2026 Conference Withdrawn Submission_

### Official Review · Reviewer_TZh4 · 2025-10-25

**Soundness:** 2
**Presentation:** 1
**Contribution:** 2
**Rating:** 2
**Confidence:** 5

**Summary:**

This work highlights that precise layer localization is crucial for activation-engineering-based steering of LLM behavior. The authors address this gap from a mechanistic interpretability perspective by introducing Edge Attribution Patching with Integrated Gradients (EAP-IG). They evaluate their proposed A2A method on two tasks: personality steering (PersonalityEdit) and alignment hacking (SafeEdit). The results show that A2A achieves performance comparable to an exhaustive grid search over layers.

**Strengths:**

- The proposed layer localization method A2A is technically sound. Analylize layer importance via the attribution graph upon input embeddings, output unembeddings, attention heads and MLP blocks is resonalble and to my knowledge is the first try in activation module  selection LLM behavior steering.
- A2A is evalualate on LLMs from two family, LLaMA2 and Qwen2.5.
- The authors provide experimental results to demonstrate the consistency on layer seletion, task accrucy and ablation on top layer number K and attribution dataset size.

**Weaknesses:**

- The presentation needs significant improvemnt. For example, the captions for Figures 2 and 3 are identical; each figure should have a distinct, informative caption.
- The paper repeatedly claims to evaluate A2A on LLaMA 2 and Qwen 2.5, but Tables 1–3 do not specify the exact model variants. Please clarify this in the experimental setup and/or table captions.
- A2A provides four metrics for layer ranking. Please explain how the metric is chosen in practice, justify the choice used in your main results, and, if possible, include a comparison to show sensitivity to the metric.
- The module-selection baselines appear incomplete. In particular, linear probing used in ITI is missing. Please include this baseline or provide a clear justification for its exclusion.

* [1] Li et al., Inference-Time Intervention: Eliciting Truthful Answers from a Language Model. In NeurIPS 2023.

**Questions:**

Please see weaknesses

---

### Official Review · Reviewer_A5Zq · 2025-10-25

**Soundness:** 2
**Presentation:** 2
**Contribution:** 2
**Rating:** 2
**Confidence:** 4

**Summary:**

This paper proposed A2A, a framework for efficiently identifying which layers in a LLM are most effective for activation steering. It constructs an attribution graph using EAP-IG to trace how internal activations contribute to specific behaviors, based on contrastive data pairs. Then aggregates edge level attributions into layer level importance scores to rank layers by influence, enabling interventions only at the top-k most impactful layers.

**Strengths:**

1. The paper reframes identifying important layers as an interpretable problem by leveraging attribution graphs, providing a principled explanation for why certain layers are effective instead of relying on empirical heuristics. And the introduction of multiple aggregation metrics (In, Out, Total, Ratio) reveals different perspectives on layer influence.
2. A2A replaces exhaustive layer wise grid search with a targeted, attribution based ranking method that reduces computation.

**Weaknesses:**

1. The experiment section lacks clarity. It is not clear which representation steering method was applied to the identified layers, nor what specific evaluation metrics were used to assess steering effectiveness. The paper would benefit from reporting quantitative measures of steering accuracy and from comparing the performance of steering only the selected layers versus steering all layers.
2. It is unclear how the attribution in Equation (1) is computed. Providing a detailed explanation of the attribution mechanism, along with intuition for its design would greatly improve it.

**Questions:**

See Weaknesses.

---

### Official Review · Reviewer_d1dy · 2025-10-31

**Soundness:** 2
**Presentation:** 3
**Contribution:** 2
**Rating:** 4
**Confidence:** 4

**Summary:**

The authors propose a way to select the most efficient layers in a transformer model using attribution patching + integrated gradients, and compare the performance gains and computational efficiency to grid-search based approaches.

**Strengths:**

-	The authors propose patching techniques instead of a grid search as the most efficient way to identify important layers for steering.
-	The authors test out their approach on multiple models and task settings, and show empirical generalization.
-	Nice complexity analysis.

**Weaknesses:**

-	The motivation is very weak, since the findings in the paper are already well known. Essentially the authors suggest that identifying important layers using causal mediation analysis is useful for steering. This is a rehashing of results from existing literature, but done in a set of specific task settings. Sadly, I didn’t really learn anything new from this paper.
-	What’s NDCG? How is it different from nDCG?
-	The authors seem unaware of the literature in causal mediation analysis, or at least don’t cite it. The framework the authors propose is essentially causal mediation analysis and it has been used in several seminal works, and shown to be both performant and computationally effective. The steering setting is not sufficiently different from these other settings – steering is essentially a “how to steer” approach once a “where to steer” site has been identified.
- No comparisons with other causal intervention approaches

**Questions:**

- The authors test out attribution patching + IG. How does this compare to activation patching, Knockout layers, adding a small perturbation to the layer in the style of the ROME paper [1]?
- How is this different from traditional causal mediation analysis approaches, besides the fact that we use CMA + steering rather than CMA + mean ablations?

---

### Official Review · Reviewer_BnWL · 2025-11-01

**Soundness:** 3
**Presentation:** 3
**Contribution:** 3
**Rating:** 6
**Confidence:** 4

**Summary:**

The paper proposes Attribution-to-Action (A2A), a computationally efficient and interpretable framework for selecting steering layers in LLMs. Instead of exhaustive layer-wise search, A2A builds attribution graphs from contrastive examples to rank layers by importance. Experiments on behavior control tasks (e.g., personality editing, jailbreak) show A2A achieves near-optimal performance with significantly fewer interventions across multiple models (LLama 2 7/13B, Qwen 2.5 7B).

**Strengths:**

- Addresses a practical challenge in activation steering: selecting the most effective layers for intervention.
- Well-grounded in mechanistic interpretability; the use of attribution graphs per layer is intuitive and well-motivated.
- Method is clearly presented with intuitive visualizations, thoughtful design choices, and empirical justifications.
- Simple, easy-to-understand approach that is significantly more efficient than exhaustive search; includes computational cost analysis.
- Offers comprehensive hyperparameter analysis (e.g., top-k selection, dataset size), with useful insights into efficient steering vector computation.

**Weaknesses:**

See Questions

**Questions:**

- Prior work [1,2,3,4] has shown that steering across all layers is effective. The paper should compare against steering on all layers as well as on top-k layers selected by A2A.
- Minor: (Line 123) duplicate citation of Hanna et al.
- Line 164–165: Why use the sum of incoming and outgoing attributions rather than the difference? Intuitively, $Out(l) - In(l)$ may better reflect layer $l$'s contribution. This is supported by the insight in lines 342–343, where A2A-In performs the worst.
- The method is only applied to activation addition (Eq. 3). The authors should also evaluate with other steering operators, such as the one used in [2].
- It's unclear how the attribution of a single layer compares to attribution aggregated across all layers. The paper should define a metric to quantify this.
- Table 4 should be moved from the appendix to the main text.

**Additional suggestions:**
- Evaluate on smaller (e.g., 2–3B) and larger (e.g., 13–14B) models, and include other model families such as Gemma to strengthen claims of generality.
- Explore the use of top-p for selecting layers.
- Consider more recent steering formulation such as [4,5].

**References:**

[1] The Hydra Effect: Emergent Self-repair in Language Model Computations
[2] Refusal in Language Models Is Mediated by a Single Direction
[3] Beyond Linear Steering: Unified Multi-Attribute Control for Language Models
[4] Angular Steering: Behavior Control via Rotation in Activation Space
[5] Controlling Language and Diffusion Models by Transporting Activations

---

### Note · Authors · 2026-01-03

I have read and agree with the venue's withdrawal policy on behalf of myself and my co-authors.